# Treatment of Secondary Dust Produced in Rotary Hearth Furnace through Alkali Leaching and Evaporation–Crystallization Processes

**Shuang Liang, Xiaoping Liang * and Qian Tang**

College of Materials Science and Engineering, Chongqing University, Chongqing 400030, China;
Shuang_Liang7@163.com (S.L.); Qian_Tang6@163.com (Q.T.)

* Correspondence: xpliang@cqu.edu.cn

**Abstract:** This paper presents the results of an experimental study on the extraction of KCl and the improvement of the zinc grade of secondary dust obtained from rotary-hearth-furnace secondary dust (RHF secondary dust) using alkali leaching ($Na_2CO_3$ solution) and evaporation–crystallization processes. The effects of the liquid–solid ratio and $Na_2CO_3$ content on the element leaching ratio in the alkali leaching process, as well as the effects of the volume–evaporation ratio and cooling temperature on KCl extraction in the evaporation–crystallization process, were investigated. The results showed that the optimum liquid–solid ratio was 6:1, and the optimum quantity of $Na_2CO_3$ was 1.5 times the basic quantity. The recovery ratio of zinc reached 95.23%, and the leaching ratio of K reached 79.01%. The experimental results of the evaporation–crystallization process demonstrated that the evaporation temperature was 80 °C, the volume evaporation ratio was 50%, the cooling temperature was 25 °C, and the mass fraction of $K_2O$ in the obtained crystals was 58.99%.

**Keywords:** secondary dust; rotary hearth furnace; alkali leaching; zinc; potassium

## 1. Introduction

Currently, energy conservation and emission reduction are strongly advocated worldwide, and different kinds of dust produced by the steel industry during the smelting process are treated to achieve environmental friendliness and resource reuse. Various methods of metal recycling are applied, including selective extraction, ion exchange and flotation, and precipitation and vacuum metallurgical technologies (among which, hydrometallurgical methods yield significant results in terms of recycling valuable elements) [1–4]. Zinc-containing dust is not only a toxic substance, but also contains valuable elements. The extraction of valuable substances from steel-containing zinc dust has attracted increasing attention from metallurgists.

China is a major producer of steel in the world. In 2019, China's crude steel output reached a record high, exceeding 900 million tons [5], and the total quantity of dust produced by steel companies is approximately 8%–12% of the steel output, in which zinc-containing dust accounts for 20%–30% of the total quantity of dust [6]. These dusts come from all aspects of iron and steel production, and mainly exist as raw material dust, sintering dust, blast furnace dust, converter dust, electric furnace dust, continuous casting, and rolled steel scale [7]. The common feature of these dusts is that they contain large quantities of iron, some of which also contain other elements, such as carbon, zinc, potassium, and sodium. From the perspectives of resource recycling, environmental protection, and sustainable development, the complete recycling of valuable dust elements in iron and steel plants should be a major focus in metallurgical resource recycling [8]. Currently, for dust containing iron and low zinc content (Zn ≤ 0.1%), iron present in the dust is recycled in the steel mills using the return

sintering method [9]. For iron-containing dust with high zinc content, the return sintering method cannot be adopted for treatment, because the circulation and enrichment of zinc will affect the sintering efficiency and reduce the life of the blast furnace; presently, it is mainly processed using the rotary kiln or rotary hearth furnace (RHF) [10,11]. Compared with the rotary kiln treatment method, the RHF treatment method is more adaptable to different kinds of dust in steelworks, and its metallization ratio and de-zincification ratio are also higher [12]. RHF treatment is considered to be an adequate method of recycling dust that contains zinc and iron. The principle of the RHF process is described in [13]: various types of dust from steel plants are mixed with carbon-containing powders to form pellets. The pellets are then reduced to iron-containing metalized pellets in the RHF at a high temperature, and the zinc and alkali metal chloride in the pellets are vaporized into flue gas to obtain the secondary dust of the RHF. The flow chart of the process is shown in Figure 1. The RHF secondary dust contains several valuable elements, such as zinc and alkali metal chloride. The recycling of RHF secondary dust is a topic of significant interest.

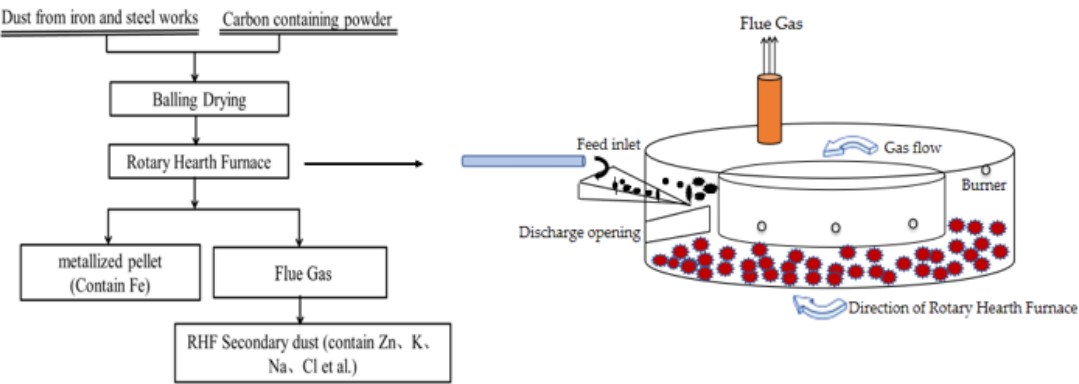

**Figure 1.** Process flow chart of RHF.

For the various types of zinc-containing dust produced in steel plants, the characteristics of the dust composition have been determined in previous studies, as shown in Table 1. In general, blast furnace dust, converter dust, and electric furnace dust contain significant quantities of iron and zinc. The alkali metal content is not high, and the zinc-containing phase is composed of $ZnFe_2O_4$ and $ZnO$. The zinc smelting waste residue has a complex phase composition, but it does not contain alkali metals. The RHF secondary dust has a low iron content, high zinc and alkali metal content, and complex zinc-containing phases. Researchers have adopted various processes in treating dust with different components. For blast furnace dust [14–17], Fu et al. [14] investigated the effect of process parameters on zinc and iron separation by reducing roasting–magnetic separation–acid leaching treatment of blast furnace dust. For electric-arc furnace dust [18–22], Tsakiridis et al. [18,19] recovered the zinc content in the form of electro-zinc plate using the sulfuric acid leach–jarosite method and removal of iron-Cyanex 272 by an extraction–electrodeposition process. Dutra et al. [22] studied the process of alkaline leaching of zinc from electric furnace dust using sodium hydroxide as the leaching agent. Converter dust [23–26] is usually used as a raw material directly or as a pelletizer in the form of balls and returned to the blast furnace. For zinc smelting waste residues [27–30], Chen [27] examined the hot-acid leaching of zinc smelting residues, reduction and purification of the leaching solution, and extraction of indium and zinc using tributyl phosphate. Turan et al. [28] investigated the process of recovering lead and zinc from the residues of the zinc smelting plant. In their study, the residue (containing 11.3% zinc and 24.6% lead) was mixed with concentrated sulfuric acid. After calcining the residue at 200 °C for 30 min, the residue was first leached with water and then leached with NaCl to recover the lead. For RHF secondary dust [31–34], Tang et al. [31] applied the water immersion–KOH precipitation–calcination process for treating the RHF secondary dust and preparing ZnO products. The method was simple and cost-effective, the composition of the raw materials was relatively simple, and the zinc-containing phases were $Zn_2SO_4$ and $ZnO$. Liu [32] studied the extraction of lead and

zinc from RHF secondary dust through the alkaline leaching–electrolysis process. Wu [33] used the normal-pressure sulfuric-acid-leaching and goethite-oxidative hydrolysis method in processes such as iron and copper removal, zinc powder replacement, and electrowinning zinc extraction, in treating secondary dust and extracting dust from a lead furnace. The RHF secondary dust used by the two researchers had high zinc contents, and the zinc-containing phase are $Zn_2SO_4$ and $ZnO$. Their studies focused on the extraction of lead and zinc. Large quantities of $NaOH$ or $H_2SO_4$ were used for the experimental tests. However, these compounds easily cause pollution and do not involve the recovery of alkali metals. Currently, most studies on steel dust have focused on blast furnace dust, converter dust, electric furnace dust, and zinc residue, and very few related studies on the treatment of RHF secondary dust have been carried out.

**Table 1.** Main chemical composition characteristics of zinc-containing dust in steel mills.

| Zinc Dust Species | Chemical Composition | Zn Range (wt%) | Fe Range (wt%) | K Range (wt%) | Na Range (wt%) | Main Phase Containing Zinc |
|---|---|---|---|---|---|---|
| Blast furnace dust [14–17] | Fe, Si, Al, K, Cl, Ca, ZnNa, C, et al. | 0–17 | 10–40 | 0.04–0.8 | 0.05–0.7 | ZnO |
| Electric furnace dust [18–22] | Fe, Zn, Si, Ca, Na, K, Cl, et al. | 2–46 | 10–45 | 0.35–2.3 | 0.5–1.8 | $ZnFe_2O_4$, ZnO |
| Converter dust [2–26] | Fe, Zn, Si, Ca, Na, K, Cl, et al. | 0–7 | 41–68 | 0.2–1 | 0.2–0.5 | $ZnFe_2O_4$, ZnO |
| Zinc smelting waste residue [27–30] | Zn, Fe, S, Pb, Cd, Ag, Si, et al. | 20–50 | 2–18 | - | - | $ZnFe_2O_4$, ZnO, $Zn_2SO_4$, ZnS, $Zn_2(SiO)_4$ |
| Secondary dust of RHF [31–34] | Zn, K, Na, Fe, Cl, Ca, Mg, et al. | 21–80 | 1.2–5 | 10–23 | 3–8 | ZnO, $ZnCl_2$, $Zn_2SO$, ZnS |

The compositions of zinc-containing dust in different production links of steel plants are different. Even zinc dust from the same production link could have differences due to the varying conditions of the raw material source, production process, equipment, and management level. The RHF secondary dust investigated in this study had a low zinc content, a complex zinc phase, water-soluble $ZnCl_2$, and water-insoluble $ZnO$. Considering that the RHF secondary dust is a special type of dust with a complex composition, harmful chlorine-containing substances and various zinc-containing valuable resources coexist in the dust simultaneously. Therefore, there is an urgent need to develop an effective treatment method that will be suitable for treating the secondary zinc dust produced in RHFs. This development will not only enrich the zinc in RHF secondary dust effectively but will also be applied in extracting the alkali metal chloride in the RHF secondary dust.

In this study, alkali leaching and evaporation–crystallization processes of the RHF secondary dust were experimentally investigated, to extract KCl and improve the zinc grade. One of the purposes of alkali leaching is to make the zinc in the RHF secondary dust in the form of $ZnCl_2$ react with $Na_2CO_3$, form precipitation, and enter the residue. Zinc in the RHF secondary dust in the form of $ZnO$ also enters the residue because it is insoluble in the $Na_2CO_3$ solution. Another purpose of alkali leaching is to make the alkali metal chloride enter the leaching solution. After RHF secondary dust is treated through alkali leaching, the main elements in the leaching solution are sodium, potassium, and chloride. KCl was extracted through the evaporation–crystallization process based on the difference in the solubility of KCl and NaCl at different temperatures. The process flow chart is depicted in Figure 2.

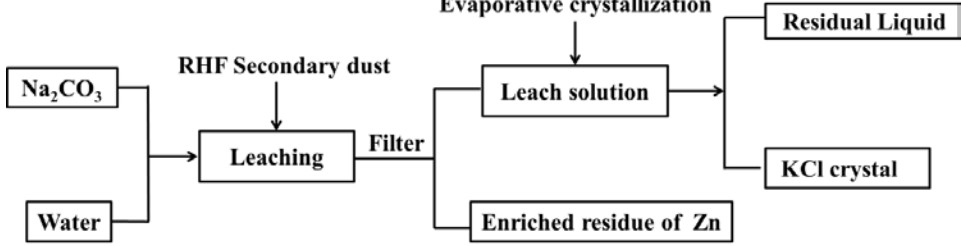

**Figure 2.** Process flow chart of Rotary Hearth Furnace secondary dust alkali-evaporation crystallization.

## 2. Experimental

### 2.1. Materials and Methods

The raw material used in the experimental test was RHF secondary dust produced in a steel plant during the steel production process (Figure 3). The concentration of elements in the RHF secondary dust was analyzed using an inductively coupled plasma emission spectrometer (ICP-OES). X-ray diffraction (XRD) was used to characterize the phase of the RHF secondary dust and leach residue. A JJ-1A electric mixer (Jintan City Experimental Instrument Factory. (Jiangsu, China)) was used for mixing. The leaching residue and leaching solution were separated through suction filtration using an SHZ-D circulating water vacuum pump (Shanghai Yikai Instrument Equipment Co., Ltd. (Shanghai, China)). A 101a-3b electric blast drying oven (Shanghai Test Instrument Factory Co., Ltd. (Shanghai, China)) was utilized for drying the leaching residue. The temperature and stirring speed were controlled using a magnetic enamel digital-display stirrer hot plate, and the pH value of the solution was measured using a pH meter.

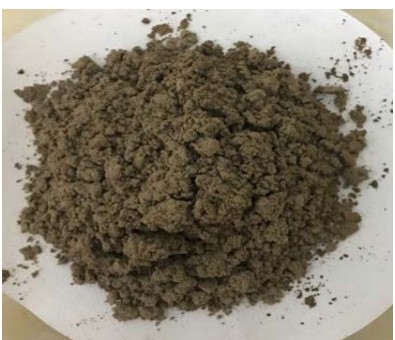

**Figure 3.** RHF secondary dust.

### 2.2. Experimental Scheme

Given the characteristics of the RHF secondary dust and the advantages and disadvantages of various leaching agents, the reason why sodium carbonate is chosen as the leaching agent is that the process is simple, and the equipment and technical requirements are not expensive and high, respectively. In addition, compared with the products made using leaching agents, such as $Na_2S$, $H_2S$, and NaOH, the purity of $Na_2CO_3$ is high.

The phase equilibrium was calculated using HYDRA-MEDUSA software [35] to determine the effect of adding sodium carbonate to the aqueous phase on the morphology of the phase in the solution. Potassium and sodium in the secondary dust of RHF could exist in the form of $K^+$ and $Na^+$, and the addition of $Na_2CO_3$ will not affect its phase. The pH of the solution system was between 8.5–11, and $Zn2^+$ in the leaching solution was converted into solid $Zn_5 (OH)_6(CO_3)_2$. $Na_2CO_3$ is a suitable leaching agent for the secondary dust of RHF. The specific calculation process is described in detail in the Appendix A.

The calculation was based on the element concentration in the water immersion solution without adding $Na_2CO_3$ and the chemical reaction formula for the precipitation reaction of ions and $Na_2CO_3$ in the water immersion solution to determine the basic addition quantity of $Na_2CO_3$ for the alkali leaching process. Based on the liquid–solid ratio (i.e., ratio of distilled water to RHF secondary dust of 100 g mass) of 2:1 in the water immersion experiments without the addition of $Na_2CO_3$, it was assumed that $Na_2CO_3$ completely reacts with the zinc, magnesium, and calcium in the water immersion solution. As the Fe content in the water immersion was very low, the reaction between Fe and $Na_2CO_3$ was ignored. The quantity of $Na_2CO_3$ to be consumed by zinc, magnesium, and calcium in the water immersion solution was calculated based on the proportion shown in chemical reaction Equations (1)–(3), respectively. Based on Table 2 and chemical reaction Equations (1)–(3), the masses of zinc, calcium, and magnesium in the water immersion solution obtained per 100 g of RHF secondary dust were 13.77, 0.58, and 0.192 g, respectively. The basic amount of Na2CO3 added per 100 g of RHF secondary dust was 24.85 g.

$$Zn^{2+} + Na_2CO_3 = ZnCO_3 \downarrow + 2Na^+ \tag{1}$$

$$Ca^{2+} + Na_2CO_3 = CaCO_3 \downarrow + 2Na^+ \tag{2}$$

$$Mg^{2+} + Na_2CO_3 = MgCO_3 \downarrow + 2Na^+ \tag{3}$$

**Table 2.** Element concentration of RHF secondary dust water immersion.

| Liquid–Solid Ratio | Element Content in Water Immersion Solution (g/L) | | | | | | |
|---|---|---|---|---|---|---|---|
| | K | Na | Cl | Zn | Fe | Ca | Mg |
| 2:1 | 55.30 | 35.79 | 193.94 | 66.21 | 6.61 | 2.77 | 0.92 |

Based on the necessary addition quantity of 24.85 g of $Na_2CO_3$, the alkali leaching experimental scheme shown in Table 3 was designed for different experimental conditions. Under the condition of the liquid–solid ratio of 2:1, the solubility of $Na_2CO_3$ was reached when the quantity of the added $Na_2CO_3$ was 1.3 times. Therefore, the maximum multiple of $Na_2CO_3$ added was 1.3 times, when the liquid–solid ratio was 2:1.

**Table 3.** Experimental scheme of RHF secondary dust saturated alkali leaching.

| Experiment Number | Liquid–Solid Ratio | Basic Addition Amount Multiple | Consumption of $Na_2CO_3$ (g) |
|---|---|---|---|
| 1 | 2:1 | 1 | 24.85 |
| 2 | 2:1 | 1.15 | 28.61 |
| 3 | 2:1 | 1.3 | 32.34 |
| 4 | 4:1 | 1 | 24.88 |
| 5 | 4:1 | 1.25 | 31.09 |
| 6 | 4:1 | 1.5 | 37.31 |
| 7 | 6:1 | 1 | 24.87 |
| 8 | 6:1 | 1.25 | 31.09 |
| 9 | 6:1 | 1.5 | 37.31 |

For the evaporation–crystallization experiment on the RHF secondary dust alkali-leaching solution, an evaporation–crystallization experimental scheme was designed. This scheme was designed to determine the effect of volume evaporation ratio and cooling temperature on the extraction of potassium from the secondary dust leaching solution of the RHF, as shown in Table 4.

**Table 4.** Experimental scheme for RHF secondary dust alkali leaching liquid evaporation.

| Influence Factor | Experimental Parameters |
|---|---|
| Evaporating temperature | 80 °C |
| Volume–evaporation ratio | 30%, 40%, 50%, 60% |
| Cooling temperature | 5 °C, 15 °C, 25 °C, 35 °C |

### 2.3. Experimental Procedure

The steps in the RHF secondary dust alkali leaching experiment were carried out as follows. Distilled water was poured into a 500 mL glass beaker, and the corresponding quantity of $Na_2CO_3$ was weighed, put in water, and stirred to ensure that the $Na_2CO_3$ was completely dissolved. Then, 100 g of the RHF secondary dust was weighed and transferred into a beaker. The mixer was adjusted to 350 rpm and stirred for 60 min. After the leaching was completed, suction filtration was used to separate the leaching residue from the leaching solution, and the pH value of the leaching solution was measured using a pH meter. Then, the leaching residue was placed in a drying box, and weighed after drying for 1 h. The potassium, sodium, chloride, zinc, calcium, magnesium, and iron in the leaching solution was measured using ICP-OES, and the phase of the residue after drying was detected using XRD.

The evaporation–crystallization experiment on the RHF secondary dust leaching solution was performed as follows. The leaching solution, under the best experimental conditions for alkali leaching, was taken and poured in a 500 mL beaker. The temperature of the magnetic enamel digital display stirrer was raised to 80 °C, and the beaker was placed on a platform for constant temperature evaporation. Then, the beaker was placed in a constant-temperature water set to cool the crystals, and the cooling time was 60 min. After the cooling and crystallization processes were completed, the solution and crystals were separated through suction filtration using a vacuum pump, and the crystals were placed in a drying box for 1 h and weighed. ICP-OES was used to detect the potassium content in the crystals and leaching solution.

(1) The calculation formula of the element leaching ratio is as follows:

$$\eta = \frac{V_L \times k_L}{M_R \times k_R} \times 10^{-6} \times 100\% \tag{4}$$

(2) The calculation formula of the zinc recovery ratio for the RHF secondary dust is as follows:

$$w = \frac{K_S \times M_S}{M_R \times k_R} \times 100\% \tag{5}$$

where $\eta$ is the element leaching ratio, %; $V_L$ is the volume of the leaching solution, mL; $K_L$ is the element concentration in the leaching solution, mg/L; $M_R$ is the quantity of the RHF secondary dust, g; $K_R$ is the element content in the RHF secondary dust, %; $W$ is the zinc recovery ratio in the RHF secondary dust, %; $K_S$ is the content of zinc in the leached residue, %; $M_S$ is the quantity of the leaching residue, g.

(3) The formula for calculating the crystallinity of the potassium element and the mass fraction of $K_2O$ in the crystals is as follows:

$$\alpha = \frac{M_C \times W_C}{V_L \times k_{L0} \times 10^{-6}} \times 100\% \tag{6}$$

$$\beta = M_C \times W_C \times \frac{94}{78} \times 100\% \tag{7}$$

where $\alpha$ is the crystallization ratio of K, %; $M_C$ is the quality of the crystal, g; $W_C$ is the potassium content in the crystal, %; $V_L$ is the volume of the leaching solution, mL; $k_{L0}$ is the potassium concentration in the leaching solution, mg/L; $\beta$ is the mass fraction of $K_2O$ in the crystal, %.

## 3. Results and Discussion

### 3.1. Characteristics of the RHF Secondary Dust

The test results on the elemental composition and content of the RHF secondary dust are presented in Table 5. The XRD results on the composition of the elements in the RHF secondary dust are shown in Figure 4. In addition to ICP-OES and XRD, chemical analysis was performed on the dust to determine the content of ZnO and $ZnCl_2$ in the RHF secondary dust further. The chemical analysis results are listed in Table 6.

**Table 5.** Composition and content of RHF secondary dust (wt%).

| Zn | K | Na | Fe | Ca | Mg | Al | Cu | Si |
|------|-------|------|------|------|------|------|------|------|
| 22.78 | 12.77 | 7.92 | 3.73 | 1.00 | 0.49 | 0.29 | 0.15 | 0.68 |

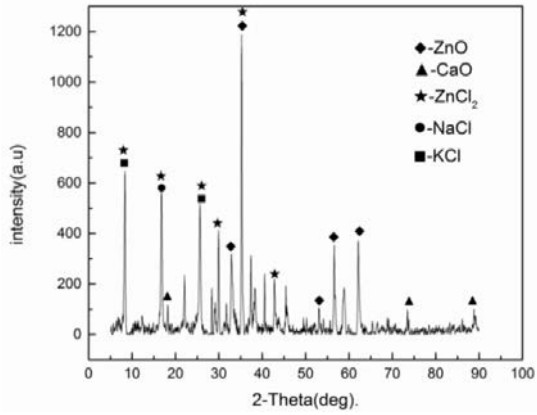

**Figure 4.** X-ray diffraction pattern of RHF secondary dust.

**Table 6.** Chemical analysis results of RHF secondary dust.

| Composition | $Fe_2O_3$ | ZnO | CaO | MgO | $SiO_2$ | Cl |
|-------------|-----------|------|------|------|---------|-------|
| content (wt%) | 1.49 | 6.84 | 0.39 | 0.24 | 0.88 | 47.22 |

From Tables 5 and 6 and Figure 4, it can be seen that the zinc content in the RHF secondary dust was 22.78%, the zinc-containing phase included ZnO and $ZnCl_2$, and the ZnO content was 6.84%. According to the calculation results for the test data, 22.78% of the RHF secondary dust contained 24% zinc in the form of ZnO and 76% zinc in the form of $ZnCl_2$. The potassium content was 12.77%, which was in the form of KCl. The sodium content was 7.92% and existed as NaCl, whereas calcium existed as CaO. No other compound was detected.

### 3.2. Effect of Technological Parameters of Alkali Leaching on Element Leaching Ratio

Nine groups of alkali-leaching experiments were performed according to the experimental scheme of Table 3 to determine the effect of the liquid–solid ratio and the quantity of added $Na_2CO_3$ on the leaching ratio of the elements. The content of the elements detected in each group of experiments is listed in Table 7. The concentration of iron was too low in the leaching solution to be detected, so it was not listed in Table 7. From the results presented in Table 7, the leaching ratio of the elements in the RHF secondary dust was calculated using formula (4), and the corresponding leaching ratios of zinc, potassium, sodium, chloride, calcium, and magnesium are shown in Figures 5–7.

**Table 7.** Test results for element content in a leaching experiment with $Na_2CO_3$.

| Experiment Number | Testing Result | | | | | | | | | |
|---|---|---|---|---|---|---|---|---|---|---|
| | Element Content in Leaching Solution (g/L) | | | | | | Element Content in Leached Residue (wt%) | | | |
| | K | Na | Cl | Zn | Ca | Mg | K | Na | Cl | Zn |
| 1 | 53.63 | 84.14 | 187.35 | 39.97 | 0.90 | 0.10 | 6.64 | 8.22 | 23.96 | 27.18 |
| 2 | 53.91 | 92.48 | 188.73 | 13.64 | 0.53 | 0.08 | 6.03 | 7.97 | 23.44 | 32.64 |
| 3 | 54.68 | 101.44 | 189.30 | 11.86 | 0.48 | 0.08 | 6.04 | 8.77 | 22.62 | 29.95 |
| 4 | 29.31 | 45.16 | 102.69 | 8.53 | 0.31 | 0.052 | 4.93 | 6.13 | 19.56 | 30.46 |
| 5 | 29.29 | 51.76 | 102.01 | 5.00 | 0.076 | 0.002 | 4.05 | 5.48 | 16.16 | 31.38 |
| 6 | 29.79 | 59.37 | 105.57 | 3.50 | 0.071 | 0.02 | 4.67 | 7.46 | 16.52 | 32.05 |
| 7 | 19.78 | 30.43 | 70.80 | 5.32 | 0.21 | 0.03 | 4.68 | 5.85 | 18.81 | 30.99 |
| 8 | 19.84 | 34.99 | 70.24 | 2.91 | 0.04 | 0.008 | 4.68 | 6.59 | 15.96 | 29.88 |
| 9 | 19.78 | 39.35 | 69.97 | 2.04 | 0.04 | 0.01 | 3.96 | 5.93 | 15.67 | 32.26 |

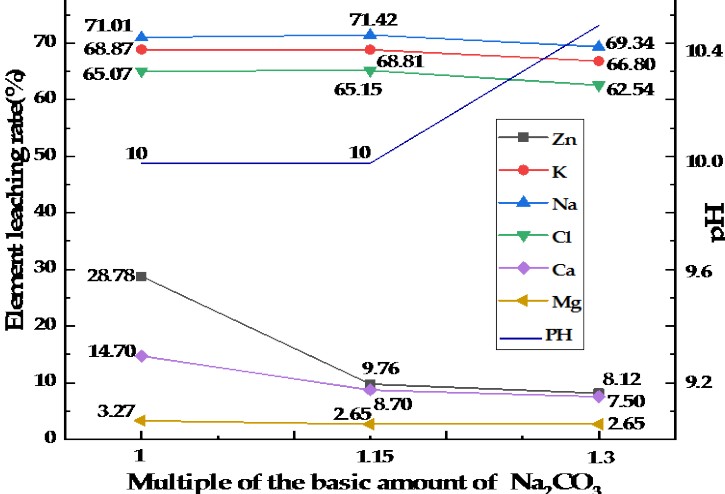

**Figure 5.** Leaching ratios of various elements with different $Na_2CO_3$ additions at a liquid–solid ratio of 2:1.

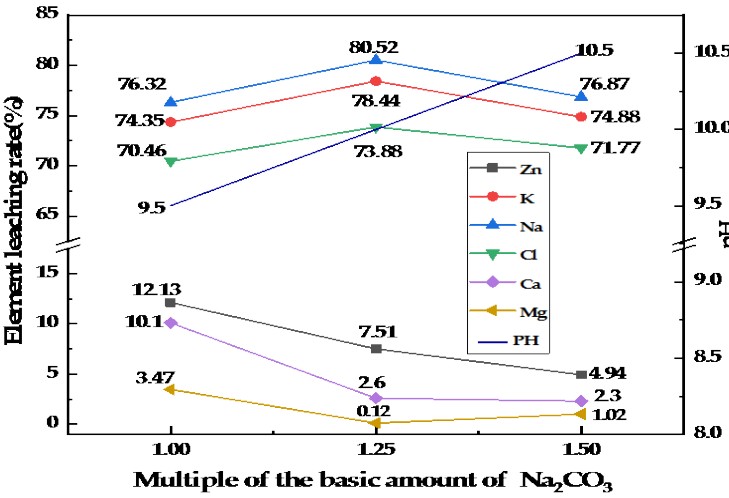

**Figure 6.** Leaching ratios of various elements with different $Na_2CO_3$ additions at liquid–solid ratio of 4:1.

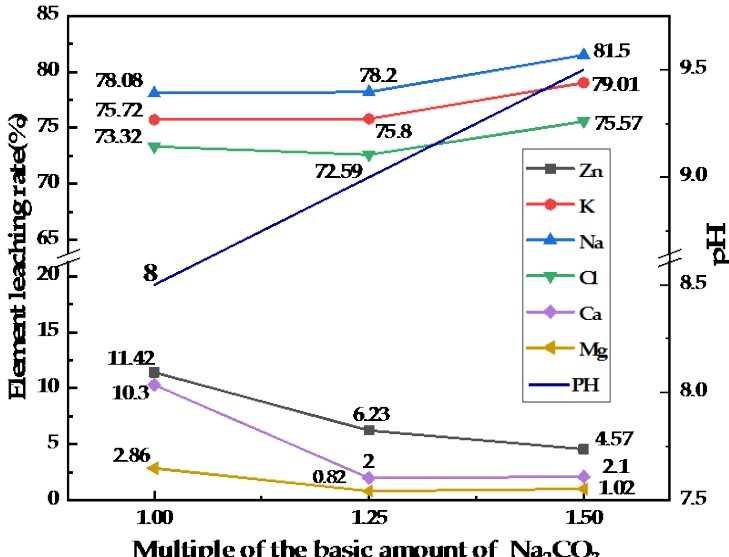

**Figure 7.** Leaching ratios of various elements with different $Na_2CO_3$ additions at liquid–solid ratio of 6:1.

It can be seen from Figures 5–7. that for the same liquid–solid ratio, the leaching ratio of zinc, calcium, and magnesium decreases with an increase in the quantity of added $Na_2CO_3$, which shows that the leaching of the $Na_2CO_3$ solution can effectively precipitate the zinc in the RHF secondary dust. At the same time, the leaching of the $Na_2CO_3$ solution can precipitate calcium and magnesium to purifying the leaching solution. When the ratio of the liquid to solid was fixed, the leaching ratios of potassium, sodium, and chloride were slightly affected by the quantity of $Na_2CO_3$.

When the $Na_2CO_3$ quantity was constant, increasing the liquid–solid ratio was beneficial to the precipitation of zinc and the leaching of potassium, sodium, and chloride in the secondary RHF dust. When the liquid–solid ratio was 2:1, the leaching ratios of potassium, sodium, and chloride were 66.80%, 69.34%, and 62.54%, respectively. When the liquid–solid ratio was 4:1, the leaching ratios of potassium, sodium, and chloride were 74.88%, 76.87%, and 71.77%, respectively. Compared with the liquid–solid ratio of 2:1, the leaching ratios of the elements increased by 7.53%–9.23%. The leaching ratios for both liquid–solid ratios improved significantly. When the liquid–solid ratio was 6:1, the leaching ratios of potassium, sodium, and chloride were 79.01%, 81.50%, and 75.57%, respectively. Compared with the liquid–solid ratio of 4:1, the yield of each element increased by 3.80%–4.63%.

In summary, increasing the liquid–solid ratio and $Na_2CO_3$ consumption is beneficial to the precipitation of zinc and the leaching of potassium, sodium, and chloride. The optimal process parameters for secondary dust alkali leaching of the rotary hearth furnace were that the liquid–solid ratio was 6:1, and $Na_2CO_3$ quantity was 1.5 times the basic quantity. The leaching ratios for zinc, potassium, and sodium were 4.57%, 79.01%, and 81.5%, respectively. The maximum leaching ratio of chloride was 75.57%.

### 3.3. Composition of Leached Residue

When the liquid–solid ratios were 2:1, 4:1, and 6:1, and the quantity of added $Na_2CO_3$ was 1, 1.25, and 1.5 times, the phase of the leaching residue from the alkali leaching test of the RHF secondary dust was detected by XRD. The phase of the leach residue was the same for different liquid–solid ratios and added $Na_2CO_3$ quantities, as shown in Figure 8. The content of the main elements in the leaching residue of the alkali leaching test of the RHF secondary dust was detected, as listed in Table 7. The comparison results of the zinc content between the leaching residue and raw material are shown in Figure 9. According to the calculation using Equations (5), the recovery ratio of zinc in the alkali leaching process was obtained, and the results are presented in Figure 10.

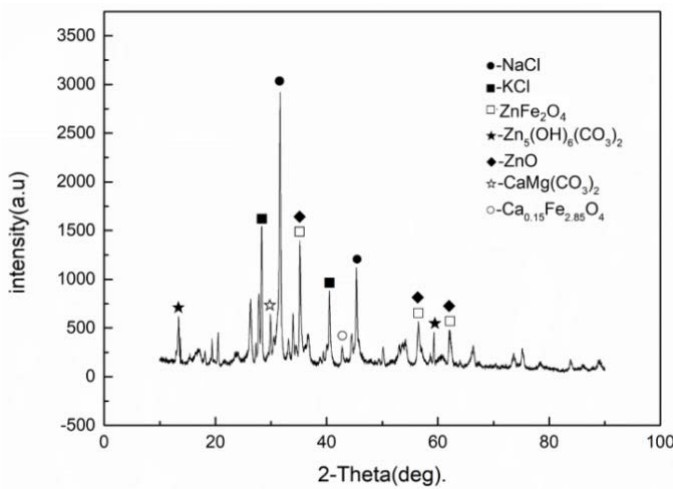

**Figure 8.** XRD diffraction pattern of leach residue (Na$_2$CO$_3$ is 1.25 times the basic addition amount, L/S 6:1).

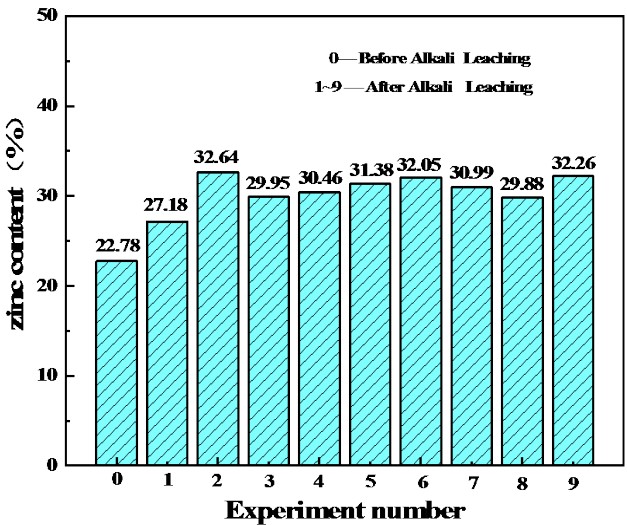

**Figure 9.** Zinc content in leaching residue for different conditions in the alkali leaching experiment.

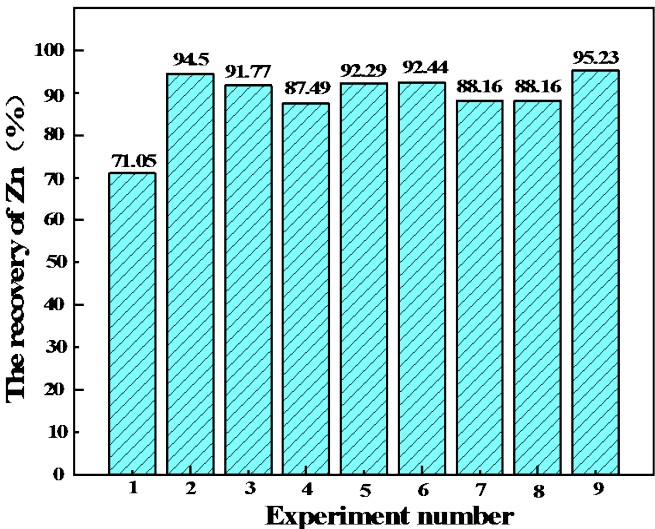

**Figure 10.** Recovery ratios of zinc in alkali leaching experiments under different conditions.

It can be seen in Figure 8 that the phase composition of the leaching residue in the alkali leaching process of the RHF secondary dust included $ZnFe_2O_4$, $Zn(OH)_6(CO3)_2$, $ZnO$, $CaMg(CO_3)_2$, and $CaO_{0.15}Fe_{2.85}O_4$. As can be observed in Figure 9, after the alkali leaching treatment, the zinc contents for different liquid–solid ratios and added $Na_2CO_3$ quantities in the experimental leaching residues were higher than those in the raw material of the RHF secondary dust. Moreover, it can be noticed in Figure 10 that under the experimental condition with a 6:1 liquid–solid ratio and 1.5 times the $Na_2CO_3$ quantity, the maximum recovery ratio of zinc reached 95.23%.

*3.4. Effect of the Evaporation–Crystallization Process Parameters on KCl Extraction*

3.4.1. Effect of Volume Evaporation Ratio on KCl Extraction

Evaporation–crystallization experiments were performed on the leaching solution obtained from the optimal alkali leaching process of the RHF secondary dust (Group 9). The experimental results for different volume evaporation ratios are listed in Table 8. Under the process conditions of evaporation temperature of 80 °C and cooling temperature of 15 °C, the volume of evaporation changed to obtain KCl crystals of different masses. According to the potassium element content in the leaching solution in Table 7 and Equations (6) and (7), the crystallization ratios of potassium and the mass fraction of $K_2O$ were calculated.

**Table 8.** Experimental results under different volume evaporation ratios.

| Volume–Evaporation Ratio (%) | Crystallization Ratio of K (%) | Crystallization Ratio of Na (%) | Crystal KCl Content (%) | Crystal K2O Mass Fraction (%) | KCl Crystal Quality (g/100 g RHF Secondary) |
|---|---|---|---|---|---|
| 30 | 20.14 | 0.13 | 95.11 | 60.00 | 4.22 |
| 40 | 45.09 | 0.88 | 94.44 | 59.58 | 8.98 |
| 50 | 52.17 | 1.07 | 92.54 | 58.38 | 10.61 |
| 60 | 71.62 | 2.67 | 60.38 | 38.09 | 22.07 |

It can be seen from Table 8 that the highest crystallization ratio of sodium in the water-immersion solution was only 2.67%, and the highest crystallization ratio of potassium was 71.62%. Hence, the evaporation–crystallization method can be used to extract KCl.

With the increase in the volume evaporation ratio, the crystallinity of potassium and the KCl crystal content steadily increased, while the mass fraction of the $K_2O$ crystals decreased. This was because, with the increase in the volume evaporation ratio, the leaching solution reached the co-saturation state of KCl and NaCl, and NaCl crystallized with KCl, reducing the $K_2O$ mass fraction of the crystal. When the volume evaporation ratio was 50%, the potassium crystallization ratio in the leaching solution and the mass of the KCl crystals were relatively maximum, so 50% was the best volume evaporation ratio for the evaporation–crystallization process.

3.4.2. Effect of Cooling Temperature on KCl Extraction

Evaporation–crystallization experiments were conducted on the leaching solution obtained from the optimal alkali leaching process of the RHF secondary dust (Group 9). The experimental results at different cooling temperatures are listed in Table 9. Under the process conditions with an evaporation temperature of 80 °C and a volume evaporation ratio of 50%, KCl crystals of various quantities can be obtained by varying the cooling temperature. According to the potassium content in the leaching solution shown in Table 7, and Equations (6) and (7), the potassium crystallization ratio and the mass fraction of $K_2O$ crystal can be calculated, respectively.

**Table 9.** Experimental results at different cooling temperatures.

| Cooling Temperature (°C) | Crystallization Ratio of K (%) | Crystallization Ratio of Na (%) | The Crystals KCl Content (%) | The Crystals K2O mass Fraction (%) | KCl Crystal Quality (g/100 g RHF Secondary) |
|---|---|---|---|---|---|
| 5 | 52.34 | 1.23 | 91.95 | 58.01 | 10.64 |
| 15 | 52.17 | 1.07 | 92.54 | 58.38 | 10.61 |
| 25 | 52.05 | 1.03 | 93.51 | 58.99 | 10.60 |
| 35 | 48.88 | 0.46 | 97.99 | 61.82 | 8.87 |

As can be noticed in Table 8, with the increase in cooling temperature, the potassium crystallization ratio and the KCl crystal content both decreased slightly, and the $K_2O$ mass fraction of the crystal increased slightly. This because, with the increase in the cooling temperature, a greater mass of the solute was required for KCl and NaCl to form a co-saturated solution, resulting in a decrease in the elemental crystallization ratio and crystal mass and a corresponding increase in the potassium purity of the crystal. Compared with 5 °C, the potassium crystallization ratio was only reduced by 0.29% at 25 °C. A temperature of 25 °C was determined as the optimal cooling temperature for the evaporative crystallization process, considering the higher energy consumption required for the cooling temperature of 5 °C.

## 4. Conclusions

In this study, the processes of alkali leaching and evaporation–crystallization of RHF secondary dust were investigated experimentally. The results demonstrate that it is feasible to apply alkali leaching and evaporation–crystallization processes to treat RHF secondary dust, which can extract KCl and improve the zinc grade. The main conclusions of this study are as follows:

The elements in RHF secondary dust are zinc, potassium, sodium, chloride, and small quantities of calcium, magnesium, iron, and other elements. Most of the zinc in the dust exists in the form of $ZnCl_2$, and a small proportion exists as ZnO. The content ratio of ZnO is approximately 8.5:1.5; potassium and sodium in the dust exist in the form of KCl and NaCl, respectively.

The results of the alkali-leaching experimental test of the RHF secondary dust show that the leaching ratios of zinc, calcium, and magnesium decrease with an increase in the quantity of $Na_2CO_3$, whereas the leaching ratio of potassium, sodium, and chlorine are less affected by the quantity of $Na_2CO_3$. The leaching ratios of zinc, calcium, and magnesium decrease with an increase in the liquid–solid ratio, while those of potassium, sodium, and chloride increase with the liquid–solid ratio. After the alkali leaching treatment is performed, the grade of zinc in the leaching residue improves significantly, and the maximum zinc recovery ratio can reach up to 95.23%.

The optimal conditions for the alkali leaching of secondary dust in RHF are as follows. When the liquid–solid ratio is 6:1, and the $Na_2CO_3$ quantity is 1.5 times the basic quantity, the recovery ratio of zinc reaches 95.23%, and the potassium leaching ratio reaches 79.01%. The optimal conditions for the alkali leaching and evaporation–crystallization processes of the RHF secondary dust solution are an evaporation temperature of 80 °C, a volume evaporation ratio of 50%, and a cooling temperature of 25 °C. Under these combined conditions, the mass fraction of potassium oxide in the obtained crystals is 58.99%.

These results provide better analysis for the recycling of RHF secondary dust containing both zinc and alkali metal chlorides. Additionally, the results provide a reference for the treatment of zinc-containing dust in steel plants with similar composition to the RHF secondary dust.

**Author Contributions:** S.L. analyzed the date and wrote the paper; X.L. conceived and designed the experiments; Q.T. performed the experiments; S.L. and X.L. revised the manuscript. All authors have read and agreed to the published version of the manuscript.

**Funding:** This research received no external funding.

**Conflicts of Interest:** The authors declare no conflict of interest.

**Appendix A**

*Analysis of Phase Equilibrium in Na₂CO₃ Leaching Solution*

The phase equilibrium was calculated using HYDRA-MEDUSA software [35] to determine the existing state of the main elements (zinc, potassium, and sodium) in the RHF secondary dust when using $Na_2CO_3$ as the leaching agent. Based on a 4:1 water immersion with a liquid–solid ratio (ratio of distilled water to 100 g of RHF secondary dust), 31.099 g of $Na_2CO_3$ was added to the water immersion solution. When using the HYDRA-MEDUSA software for calculation, the ion concentration in the secondary leaching solution of the RHF was obtained (Table A1), and the concentrations of $CO_3^{2-}$ and $Na^+$ added to the leaching solution were 0.586774 and 0.293387 mol, respectively. Figure A1 shows the effect of the leaching system added with sodium carbonate on the phase equilibrium of $Zn^{2+}$, $K^+$, and $Na^+$.

**Table A1.** Ion concentration in RHF secondary dust water immersion solution.

| Liquid–Solid Ratio | Ion Concentration in Leaching Solution (mol/L) | | | | | | |
|---|---|---|---|---|---|---|---|
| | $K^+$ | $Na^+$ | $Zn^{2+}$ | $Fe^{2+}$ | $Ca^{2+}$ | $Mg^{2+}$ | $Cl^-$ |
| 4:1 | 0.72 | 2.18 | 0.54 | 0.06 | 0.03 | 0.02 | 2.81 |

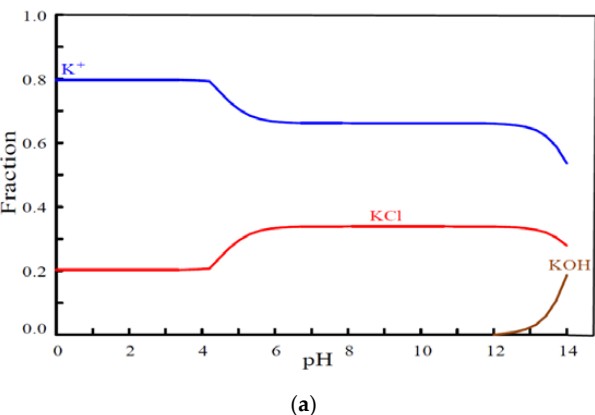

**(a)**

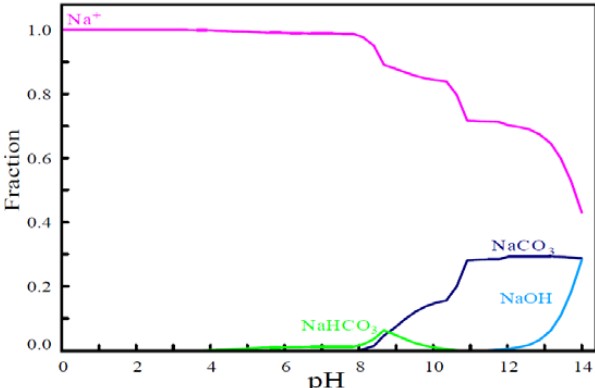

**(b)**

**Figure A1.** *Cont.*

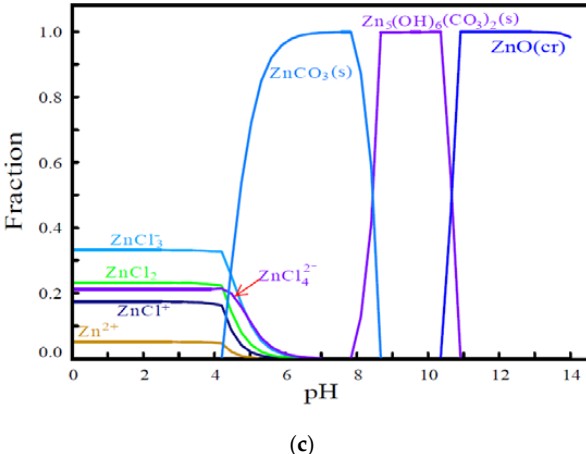

(**c**)

**Figure A1.** Effect of $Na_2CO_3$ addition on $K^+$ (**a**), $Na^+$ (**b**), and $Zn^{2+}$ (**c**) phase equilibria in leaching solution.

It can be observed in Figure A1 that potassium and sodium in the RHF secondary dust still exist in the form of $K^+$ and $Na^+$, and the addition of $Na_2CO_3$ will not affect its morphology. When the pH of the solution system ranged from 0–5, $Zn^{2+}$ contained in the leaching solution was converted to $ZnCl_3^-$. When the pH of the solution system ranged from 5–8.5, $Zn^{2+}$ in the leaching solution was converted to solid $ZnCO_3$. When the pH of the solution system was within the range of 8.5–11, $Zn^{2+}$ in the leaching solution was converted into solid $Zn_5(OH)_6(CO_3)_2$. When the pH of the solution system ranged from 11–14, $Zn^{2+}$ in the leaching solution was transformed into crystalline ZnO. Theoretically, after the RHF secondary dust undergoes alkali leaching treatment, if the pH of the solution system is between 8.5 and 11 the leaching residue containing $Zn_5(OH)_6(CO_3)_2$ and the leaching solution containing alkali metal chloride can be obtained.

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
