# Peer review of "Treatment of Secondary Dust Produced in Rotary Hearth Furnace through Alkali Leaching and Evaporation–Crystallization Processes"

_processes, doi:10.3390/pr8040396_

Round 1

Reviewer 1 Report

The paper of Liang et al. discusses the leaching with Na2CO3 of a secondary dust originating from processing of primary steel mill dust in a rotary hearth furnace. The paper discuss some empirical data presented almost as a technical report with little scientific novelty. However, the reuse of secondary resources for metal recovery is a growing interesting topic. Overall the paper can be published but the authors should address at least the following shortcomings:

The references needs to consider the global state of the art on the topic. Most of the papers cited are from local authors and some are not accessible to the broader public English needs to be revised, especially in the abstract but also throughout the text. For example the word "rate" is misused Figure 4 is trivial and can be just an appendix/supplementary rather than a subject of discussion Numbers reported in ppm are almost impossible to read. An extraction percentage would be easier to understand. I may have missed it but I did not understand how the Cl was determined 

Author Response

Dear Editor,

Thank you for your letter and for the reviewers’ comments concerning our manuscript entitled “Study on Treatment of Secondary Dust in Rotary Hearth Furnace by Alkali Leaching—Evaporation-Crystallization”. The comments are valuable and helpful for revising and improving our paper. We have studied comments carefully and made clarification which we hope meet with approval. Revised portion are marked in red in the revised paper. The detailed responses to the reviewer’s comments are presented as follows.

Reviewer 2 Report

1)        An image or drawing of the RHF rotary hearth furnace and an image of the obtained powders should be added to the introduction.
2)        The authors should explain why they have selected Na2CO3 for the lixivation of KCl and Zn
3)        The balanced reaction of ZnCl2 and Na2CO3 should be presented (after table 1)
4)        More details about the extraction of KCl should be added
5)        In chapter 2.2, the authors obtain 24.847 g of Na2CO3 for 100 g of RHF poder. Justify that quantity.
6)        Description of figures 5, 6 and 7 (pages 9 and 10) seems confusing. For a better understanding, the graphs regarding a constant percentage of Na2CO3 (variable liquid/solid ratio) and a constant liquid/solid ratio (variable Na2CO3 percentage) should be separated.
7)        Data contained in table 8 seems to disagree with the text that follows that table. The authors indicate the máximum yield is obtained with a 50%, but the table indicates other thing.

Author Response

(The authors gave the same response as above.)

Reviewer 3 Report

Dear Authors,

Your paper titled ”Study on Treatment of Secondary Dust in Rotary Hearth Furnace by Alkali Leaching—Evaporation Crystallization” concerns an interesting subject. How to increase the recycling rate from residue streams from iron and steelmaking. However, the paper contains several mistakes that has to be addressed.

General comments:

1) Please check and correct the language.

2) All references in the introduction are from 2000 and later. Please also include what has been done earlier

3) Section 2.1 are missing information on how the chemical analysis have been performed and what equipment that was used. That’s including the XRD-analysis. Which equipment and what database for extracting which phases correlates to the peaks.

4) Generally too many decimals

5) Some kind of thermodynamic calculations has been performed however, there is no information on what kind of data that has been used. This must be included.

6) The authors are talking about a leach slag. Please change to leach residue.

Author Response

(The authors gave the same response as above.)

Reviewer 4 Report

Thank the authors for interesting research, but the paper contains a lot of design drawbacks and missing information. 

I've tried to highlight all possible drawbacks, but definitely one more revisions will be essential. 

Author Response

(The authors gave the same response as above.)

Round 2

Reviewer 1 Report

The authors of this manuscript have implemented some changes with respect to the original version. However, the fundamental objections raised by the several referees have been met only partially. The manuscript does not flow nicely and the novelty of the data presented is in my opinion very little if any at all. I recommend this manuscript is not published. 

Author Response

Dear reviewer,

Thank you very much for your letter. I apologize for the ignorance of the first reply and the imperfections in the revision of the paper. Because English is not my native language, I use it incorrectly in many places. I sincerely apologize again and hope to get your forgiveness.

The comments are valuable and helpful for revising and improving our paper. We have revised the paper and responses again which we hope meet with approval. Revised portion are marked in red in the revised paper. The detailed responses to the reviewer’s comments are presented as attachment. At present, the epidemic situation of new crown pneumonia is serious, please pay attention to precautions and wish you have good health.

please see Attachment.

Reviewer 3 Report

First of all, I want to thank the authors for their effort in modifying the paper according to reviewers' comments and suggestions. I think that current version of the paper has improved, although more changes are necessary.

General comments:

Check the language. There are still several mistakes in the text like missing space after end of sentence, capital letter after comma etc.

Check the numbering of references.

Check numbering of tables and figures and the cross-referencing in the text

Detailed comments:

Section 1 Introduction:

  • In figure 1, the term smoke is used while flue gas is used in the text. Please use the same term.
  • It is not clear from the introduction why the evaporation and crystallisation part is included. Please motivate why that is important for the current work and what is new.

Section 2 Experimental

  • Include information on what type of XRD equipment that has been used together with the settings
  • In subsection 2.2: The text section describing the calculation procedure for Na2CO3 addition is not clear. Please elaborate the text. Is the data in table 2 a result of calculations described in appendix? Please explain the meaning of “water immersion solution”?
  • In subsection 2.2: It is not clear how the data in table 2 is determined?
  • In subsection 2.3, the formula for calculating the “leaching rate” is included. In my opinion, the formula expresses a leaching yield not a rate since there is no time factor involved. Please change and update the rest of the text to correspond to yield.
  • Include how the chemical analysis is performed that is referred to in section 3.1

Section 3 Results and discussion

  • From the text, it is not clear on how the content of ZnO is determined that is stated in table 5. Please include the information.
  • Include in the caption for table 5 that ZnO is part of the total Zn content.
  • According to table 5, the Fe content is 3.73% but in the text is says that Na is 3.73%. Please correct.
  • In the text below table 5, it is referred to table 2 and figure 3. Is this correct or should it be table 5 and figure 4?

Appendix

  • The appendix is not referred to in the text.
  • I don’t understand why this section has been moved from the previous position to the appendix. Please motivate.
  • Include more information regarding the thermodynamic calculations performed. Databases used etc.

Author Response

(The authors gave the same response as above.)

Reviewer 4 Report

not all suggestions were taken into account. (missing methods of determination of the metals)

Still remain a problem with using of the space (it''s lacking in many places)

Author Response

(The authors gave the same response as above.)

Round 3

Reviewer 1 Report

This is the third round of review of this manuscript, which has been improved substantially from the original draft. Overall, I did not change my assessment of this manuscript as an "average" piece. In my opinion the data and finding presented are not particularly significant. However, the manuscript can be published in the present form.   

Reviewer 3 Report

Dear Authors,

First of all, I want to thank the authors for improving the paper and taking the comments from the reviewers into consideration. My recommendation to the Editor is to accept the paper.

All the best